# MCP-Persona: Benchmarking LLM Agents on Real-World Personal Applications via Environment Simulation

**Wenhao Wang** [† * 1 2]  **Peizhi Niu** [* 1 3]  **Gongyi Zou** [* 1 4]  **Xiyuan Yang** [* 1]  **Jingxing Wang** [* 1]
**Haoting Shi** [1]  **Yaxin Du** [1]  **Jingyi Chai** [1]  **Xianghe Pang** [1]  **Shuo Tang** [1]  **Yanfeng Wang** [5 1]  **Siheng Chen** [1 5]

## Abstract

The Model Context Protocol (MCP) has emerged as a transformative standard for connecting large language models (LLMs) with external data sources and tools, and has been rapidly adopted across personal applications and development platforms. However, existing benchmarks predominantly focus on generic information-seeking tools and fail to capture the practical challenges posed by personal social applications, where tools interact with individual accounts or local databases. To bridge this critical gap, we introduce MCP-Persona, the first benchmark specifically designed for evaluating agent performance on real-world, personalized MCP tools. MCP-Persona encompasses a diverse set of widely-used applications, ranging from social media platforms like Reddit and Xiaohongshu (Rednote) to enterprise collaboration suites such as Lark (Feishu) and Slack. Our extensive experiments on various state-of-the-art (SOTA) agents demonstrate their significant struggles with personalized tool use, thereby highlighting the benchmark's crucial role in identifying and addressing these limitations. MCP-Persona is publicly available at https://github.com/wwh0411/MCP-Persona

## 1. Introduction

The Model Context Protocol (MCP) has recently emerged as a foundational abstraction for connecting large language models (LLMs) with external tools and data sources, marking an important step toward practical, tool-augmented intelligence (Hasan et al., 2025; Guo et al., 2025). In parallel, the rise of personalized AI, fueled by a shift to user-centric and on-device computing, is redefining intelligent agents. Tech giants like Apple and Alibaba are embedding LLMs into mobile assistants for daily activities. Meanwhile new frameworks like Anthropic's Skills (Anthropic, 2025) and the explosively growing OpenClaw (OpenClaw, 2025) ecosystem empower users to create custom, task-specific agents. This signals a clear transition from general-purpose assistants to deeply personalized, skill-driven counterparts.

Despite the growing importance of personalization, the evaluation of intelligent agents on real-world personalized applications and tools remains largely underexplored (Yin et al., 2025; Jia et al., 2025; Chen et al., 2025). Existing research on tool use or MCP frameworks (Liu et al., 2025; Fan et al., 2025) primarily focuses on generic tool orchestration and workflow optimization, while largely overlooking the challenges posed by personalized tools that are tightly coupled with individual user accounts, preferences, and historical behaviors. This gap is particularly problematic given that many high-impact applications of MCP, such as social media, file management, and consumer-facing automation, are inherently personalized by design. The limited progress in this direction stems from several fundamental challenges:

1. First, realistic deployment of personalized MCP servers typically requires access to private user data and substantial human effort for environment setup, making large-scale experimentation costly and difficult to reproduce.
2. Second, privacy concerns and operational security constraints significantly restrict the collection, sharing, and reuse of personal contexts, hindering the construction of open-source benchmarks.
3. Finally, maintaining a stable and executable environment that can reliably simulate numerous heterogeneous users, while offering controlled and fair evaluation for research purposes, poses non-trivial technical challenges.

Together, these obstacles have slowed systematic research on personalized MCP agents, underscoring the need for new paradigms and infrastructure that can faithfully capture real-

---

*Equal contribution [†] Project lead [1]Multi-Agent Governance & Intelligence Crew (MAGIC), Shanghai Jiao Tong University, Shanghai, China [2]Department of Computer Science and Technology, Zhejiang University, Hangzhou, China [3]University of Illinois at Urbana-Champaign, Illinois, USA [4]University of Oxford, Oxford, UK [5]Shanghai AI Laboratory, Shanghai, China. Correspondence to: Siheng Chen <sihengc@sjtu.edu.cn>, Yanfeng Wang <wangyanfeng@sjtu.edu.cn>.

*Proceedings of the 43$^{rd}$ International Conference on Machine Learning*, Seoul, South Korea. PMLR 306, 2026. Copyright 2026 by the author(s).

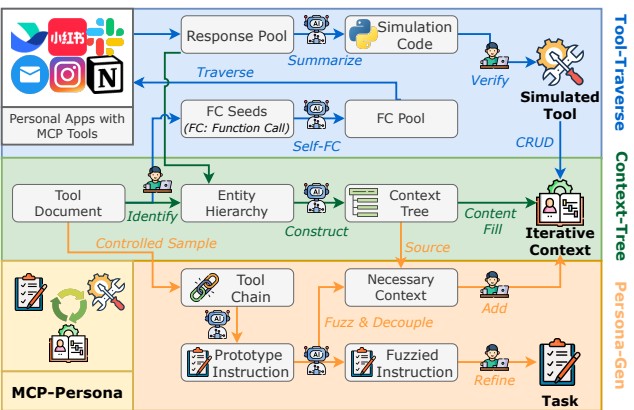

*Figure 1.* System overview of MCP-Persona, which is built upon the interaction of Tools , Contexts , and Tasks . For each component, we introduce a dedicated method, described in detail as *Tool-Traverse* (§3.1), *Context-Tree* (§3.2), and *Persona-Gen* (§3.3).

world personalization while preserving user privacy and evaluation fairness.

To address the aforementioned challenges, we introduce **MCP-Persona**, the first evaluation platform tailored for tool-enhanced agents operating over real-world personalized applications (e.g., Slack, Rednote and Instagram). As illustrated in Figure 1, MCP-Persona is built upon the interaction of three environment components: tools, contexts (user profiles) and tasks.

Specifically, for tool simulation, we propose **Tool-Traverse**, a traverse-then-simulate paradigm. After collecting a diverse set of real-world personalized MCP servers, we first manually deploy them with sandbox environments and accounts. Building upon a set of human-curated seed function calls (FCs), we introduce Self-FC, a technique inspired by Self-Instruct (Wang et al., 2023), to expand the coverage and diversity of authentic function calls. Using this enlarged FC pool, we systematically traverse the MCP servers to collect a wide range of possible response outputs. Based on these observed responses, we summarize the underlying interaction patterns and subsequently generate the executable logic for our simulated tools in the form of Python code.

For context construction, we propose **Context-Tree**, a method that simulates user contexts of an application as a structured tree. The process begins with the tool documents and FC seeds, from which we manually define a hierarchical context structure for each server. For example, in Lark, the hierarchy is User→Calendar→Event. By mapping the relationships between all tool parameters, we construct a preliminary context-tree that naturally includes redundant sub-nodes. We populate the context-tree by assigning values to each node. To maximize realism, we prioritize authentic online content (e.g., Xiaohongshu posts) when available, while sensitive fields such as phone numbers are replaced

with fakes to preserve privacy. Once populated, a context instance is supplied to the simulated tools, enabling stateful operations such as creating, modifying, and deleting entities within the environment.

For task generation, we introduce **Persona-Gen**, a two-stage pipeline involving both automated synthesis and manual refinement. The process starts with automatically generating prototype instructions from high-quality tool chains, which are sampled under the constraints of dependency, diversity, and realism. To transform these abstract prototypes into realistic tasks, we then inject rich context from our context-trees and intentionally obfuscate the instructions to mimic the ambiguity of real-world user requests. Subsequently, human annotators review each task to verify its alignment with the tool chain and refine it by adding necessary contextual details. This rigorous process yields our final set of 173 high-quality, human-verified personalized tasks.

We conduct extensive experiments on MCP-Persona to examine the performances of diverse LLM agents. The results exhibit several key findings: (1) Even SOTA agents like GPT-5 still fall short in some personalized tasks with unseen tools. They have trouble acquiring necessary information embedded in the environment that is not explicitly exhibited in the instruction. (2) By traversing authentic function calls for operation-dominated MCP servers, our simulation method yields highly accurate prediction with only limited manual efforts. (3) Incorporating dedicated skills for specific applications can improve performance, but a noticeable gap still remains before reaching satisfactory results. To summarize, our contributions are as follows:

1. We propose Tool-Traverse, a novel and rigorous simulation method designed to faithfully replicate the functionalities of real-world personal MCP tools, which cannot be openly evaluated due to account-binding requirements.
2. We construct MCP-Persona, a new benchmark comprising 173 human-verified tasks spanning four representative personalized scenarios, addressing the critical user demand for enhanced agents in communication and collaboration applications.
3. We conduct extensive experiments on over ten SOTA models and reveal their significant limitations in personalized tool usage, particularly in terms of underexploration of tool functionalities and the failure to discover information embedded within the environment.

## 2. Related Work

### 2.1. Personalized Agents in Real-World Applications

Personalized agents have rapidly proliferated into widely deployed products, forming an emerging ecosystem that supports individualized, localized, and everyday user needs. The Skills framework (Anthropic, 2025) introduced by An-

*Table 1.* Comparison of MCP-related or personalized benchmarks. MCP-Persona covers the broadest personal application domain.

| Benchmark | Real-World | Personal Context | Application Domain Coverage | | | |
|---|---|---|---|---|---|---|
| | | | Social Media | Collaboration Platform | Email | Content Management |
| AppWorld (Trivedi et al., 2024) | ✗ | ✓ | ✗ | ✗ | ✓ | ✓ |
| PersonaBench (Tan et al., 2025) | ✗ | ✓ | ✗ | ✗ | ✓ | ✓ |
| InfoMosaic-Bench (Du et al., 2025) | ✓ | ✗ | ✗ | ✗ | ✗ | ✗ |
| MCP-Universe (Luo et al., 2025) | ✓ | ✗ | ✗ | ✗ | ✗ | ✓ |
| TOOLATHLON (Li et al., 2025) | ✓ | ✗ | ✗ | ✗ | ✓ | ✓ |
| MCP-Persona (Ours) | ✓ | ✓ | ✓ | ✓ | ✓ | ✓ |

thropic exposes a modular and extensible mechanism for equipping agents with user-specific capabilities, allowing developers and end users to compose customized skills tailored to localized environments and diverse real-world demands. In the GUI domain (Wang et al., 2026; 2025b), Doubao Phone integrates on-device intelligence to deliver deeply personalized mobile experiences, enabling users to interact with agents that supports daily activities such as communication, shopping and food delivery. Very recently, OpenClaw (OpenClaw, 2025) has caught significant attention. It focuses on personalized data and task management, providing agent-driven assistance for individual workflows and personal organization, thereby demonstrating how agents can be seamlessly integrated into personal scenarios with a high degree of automation. Together, these examples illustrate a broader trend in which personalized agents are increasingly integrated into consumer-facing products and application ecosystems, emphasizing adaptability, locality, and alignment with everyday human activities.

## 2.2. Research on Real-World Tool Agents

While the success of personalized tool-augmented agents have stimulated substantial community interest, existing research rarely focus on real-world personalized scenarios, particularly those involving social media platforms and enterprise collaboration systems. They are confronted with three following limitations: (1) First, most existing tool-use benchmarks and datasets primarily target generic search-oriented tools, tasks and simplified scenarios (Luo et al., 2025; Xu et al., 2025; Fan et al., 2025), which fail to capture the complexity and personalization characteristics of real-world stateful applications. (2) Second, although a few benchmarks attempt to incorporate personalization (Trivedi et al., 2024; Tan et al., 2025), they dominantly rely on synthetic tools. For instance, Tau-Bench (Yao et al., 2024) serves as the first tool-agent-user benchmark, but it relies exclusively on synthetic airline and retail tools, whose distributions may substantially deviate from those of real-world systems. (3) Third, the most recent personalized benchmark, ToolAthlon (Li et al., 2025), provides a comprehensive evaluation suite with real-world Notion and Email tools. Never-

theless, due to the intrinsic difficulty of account-associated and permission-sensitive environments, ToolAthlon does not support task evaluation on social media platforms or enterprise collaboration tools, which are among the most widely used applications in everyday scenarios.

The gap is particularly concerning given the critical role of individualized demands in real-world agent usage. To bridge this gap, we simulate real-world tools from popular applications (e.g., Instagram and Lark) and introduce MCP-Persona, the first benchmark explicitly designed for personalized tool usage in social communication scenarios.

## 3. Methodology

### 3.1. *Tool-Traverse*: Simulating Stateful MCP Servers by Traversing Authentic Tool Calls and Responses

**MCP Server and Tool Collection.** To build a comprehensive benchmark that bridges the gap between generic and personalized tool use, we employ a hybrid collection strategy combining rigorous manual curation with automated discovery. We focus our primary data collection efforts on high-value, personalized applications that require authentication and state management. We manually curated a diverse set of 12 MCP servers spanning critical categories including enterprise collaboration platforms (e.g., Lark), social media (e.g., Instagram), and content management (e.g., Notion). To simplify testing in real-world environments, we standardize the deployment infrastructure and unify authentication by provisioning dedicated test accounts and pre-configured tokens, ensuring a stable execution environment. In parallel, we utilize the MCP-Flow framework (Wang et al., 2025a) and build an automated pipeline to harvest information-seeking servers, which are typically stateless or free of authentication, enabling discovery and deployment from open-source repositories without manual configuration. Server details are attached in Table 8.

### 3.1.1. AUTHENTIC FUNCTION CALL TRAVERSAL

To construct a high-fidelity simulator, we must first establish a function call pool that captures the authentic behavioral

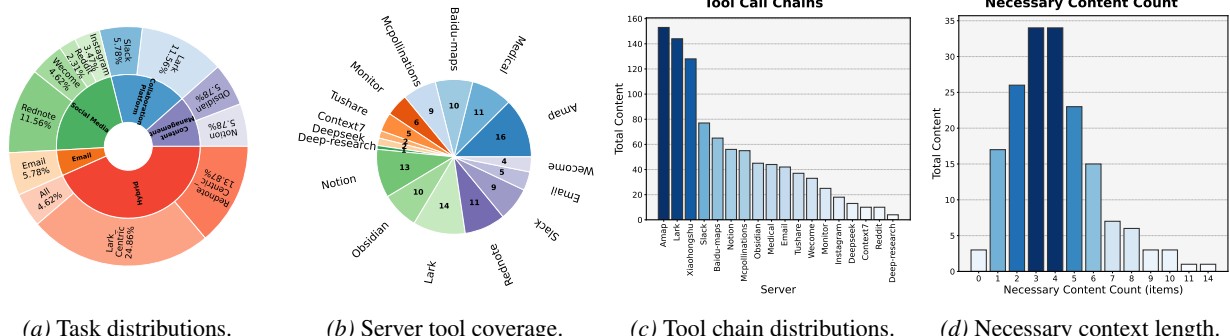

*(a)* Task distributions.   *(b)* Server tool coverage.   *(c)* Tool chain distributions.   *(d)* Necessary context length.

*Figure 2.* Dataset and tool statistics of MCP-Persona. MCP-Persona comprises a total of 24 MCP servers including 12 personalized servers. It encompasses tasks for both single-server and cross-server (hybrid) scenarios, featuring diverse tool chain and personal context distributions to ensure comprehensive evaluation.

distribution of MCP servers. Relying solely on static tools' documentation is insufficient, as it often fails to document implicit constraints and error handling logic. We therefore propose a **Tool Traversal** paradigm that systematically probes real MCP servers to map both successful operation manifolds and decision boundaries for failure cases.

**Bootstrapping with Successful FC Seeds.** For every tool $t$ within a collected MCP server set $\mathcal{T}$, we first perform positive traversal to record nominal behaviors. We parse the tool's input schema $d_t$ to identify required parameters and data types. Human annotators construct valid seed function calls $x_{seed}$ that satisfy semantic constraints (e.g., ensuring a generic ID corresponds to an existing entity). We execute these calls against the live server and record the interaction tuple $(t, x_{seed}, y_{seed}, \tau)$, where $y_{seed}$ is the authentic response and $\tau$ represents execution metadata. This establishes the baseline for correct tool functionality.

**Augmenting FC Pool by Adversarial Failure Induction.** A robust simulator must accurately replicate how servers handle invalid requests. To capture these error modes, we implement an LLM-driven adversarial call generation pipeline. We model the input call space $\mathcal{X}$ and systematically perturb valid seed inputs $x_{seed}$ to generate a set of invalid inputs $\mathcal{X}_{fail} = \{x'_1, x'_2, ..., x'_k\}$. The generation process is guided by a taxonomy of error types, ensuring coverage across: (1) *Type Mismatches*, violating primitive type constraints; (2) *Schema Violations*, omitting required fields; (3) *Boundary Conditions*, exceeding numerical ranges; and (4) *Semantic Conflicts*, where parameter combinations are syntactically valid but logically contradictory. The generated inputs are also executed against the live server. To ensure data quality, we employ a secondary LLM as a response discriminator to determine whether a tool call correctly identifies a corresponding error (returning a structured error message) or fails unexpectedly. Only verified function calls are added to the pool $\mathcal{P}_t = \{(t, x_i, y_i, \tau_i)\}_{i=1}^N$, ensuring the simulator learns to replicate specific error modes rather

than generic failures of the target MCP tool.

### 3.1.2. EXECUTABLE CODE-BASED TOOL SIMULATION

Leveraging the function call pool, we further construct a fully executable local replica of the target MCP tools by using LLMs to summarize their underlying processing logic. A critical distinction of our approach is the **Code-as-Simulation** paradigm: rather than relying on manual rule-coding or static mock responses, our pipeline employs LLMs to autonomously synthesize executable Python files that serve as the simulation engines. This approach ensures scalability across diverse tools without human intervention.

**Dynamic Context Handler.** As described in Section 3.2, real-world tools operate on contexts with hierarchical data structures. Based on this schema, we facilitate the LLM summarizer to autonomously writes a *Context Handler* module. This generated script provides a standardized, code-level interface for entity manipulation (i.e., load, save, query, and modify), allowing the final simulator to persist state changes across multiple interaction turns automatically.

**Code-Based Simulation Kernels.** Encapsulated within an individual Python script, the core logic for each simulated tool is synthesized from a pool of function calls that captures both successful and failed execution pathways. Specifically, the generation process is conditioned on: (1) the tool's input schema; (2) the collected behavioral traces; and (3) the generated context handler APIs. Based on this information, the LLM is prompted to write a complete Python code implementation, $K_t$, that acts as the transition function

$$f_t : (\mathcal{C}_{current}, x) \rightarrow (\mathcal{C}_{new}, y),$$

where $x$ and $y$ denote tool call and responses respectively. This generated function includes rigorous logic to validate inputs, check entity existence against the dynamic context (e.g., ensuring a 'calendar_id' exists before adding an event), and return appropriate responses. By training the LLM to write codes that handle specific failure modes identified in

the traversal phase, the resulting simulation kernels accurately reproduce the exact decision boundaries and error messages of the live servers. These simulators are then executed in a sandbox environment to serve user requests.

### 3.2. *Context-Tree*: Constructing MCP Server Contexts via a Tree Hierarchy of User Profiles

To enable a high-fidelity simulation, a structured and mutable context is required to support multi-turn, stateful tool execution and personalized task synthesis. Therefore, we construct a server-level context $\mathcal{C}$ using a tree hierarchy, which can interact with the dynamic context handler (Section 3.1) and the task generator (Section 3.3).

**Entity and Hierarchy Identification.** Each MCP server hosts a distinct set of entities and content structures (e.g., Slack functions through groups while Xiaohongshu involves posts). To unify them under a consistent processing pipeline, we propose a tree structure that is both well-organized and easily extensible, supporting dynamic interactions. The hierarchy of this context-tree serves as the foundation for our context construction. Specifically, for each application server, the server-specific context hierarchy defines the entity types, their associated fields (with coarse constraints), and the ownership or reference relationships among them. The hierarchy is derived by aggregating fields in the tool-call pool $\mathcal{P}_t$ collected in Section 3.1.1, and aligning repeated object patterns across tools to define entity types and relations. To guarantee robustness, we build the hierarchy via a human-in-the-loop annotation: annotators (1) consolidate entity types by grouping tools operating on the same objects, (2) aggregate fields to identify keys, references, and coarse constraints, and (3) define a hierarchy for organizing entities, typically rooted at `User` in personalized settings.

**Context-Tree Construction.** Given the tree hierarchy, we materialize a user-rooted context in which nodes correspond to typed entities and edges follow the ownership/containment relations defined in the hierarchy (e.g., `User`→`Chat`→`Message`, `User`→`Calendar`→`Event`). For each parent entity, its child entities of the same type are stored as an identifier-indexed map $\{\text{id} \rightarrow \text{obj}\}$, which ensures uniqueness and enables efficient lookup and updates. Non-containment relations are encoded via identifier references (foreign keys) rather than duplicating full objects. If different tools expose different fields of the same entity, we match by the entity ID and incorporate newly observed fields into the existing entry, keeping the context consistent across turns.

**Content Sourcing and Generation.** To populate each server-specific context-tree, we first employ an LLM to automatically assign one of four generation methods to every field in the context hierarchy. (1) *Enumerate*: constrained sampling from a fixed set; (2) *Free-Form*: LLM-conditioned

*Table 2.* Four content generation methods with example fields.

| Method | Example |
|---|---|
| *Enumerate* | `"iplocation": ["Beijing", "Shanghai"...]` |
| *Free-Form* | `"channel_name": "MCP-Flow group"` |
| *Random* | `"chat_id": "oc_50df962af-41184f144d3ebf61b0c7571"` |
| *Authentic* | `"desc": "I wonder what handwritten English looks like to a native speaker?"` (Real Rednote Post) |

free-form generation; (3) *Random*: random generation from an LLM-synthesized generator, prompted with an example and lightweight constraints (e.g., required prefixes or formats); or (4) *Authentic*: seeding with sanitized authentic text from text-heavy platforms (e.g., Rednote) that preserves realism while masking user information such as nicknames. Examples of different methods are provided in Table 2.

After the contents are filled in the context-tree, we then perform cross-entity linking. For entities that require references to others (e.g., chat members), we form links by sampling from the already generated contents and inserting the corresponding identifiers, establishing consistent relational structure.

### 3.3. *Persona-Gen*: Personalized Task Generation with Tool Invocation Chain and Instruction Fuzzification

Based on the developed simulation environments, we further introduce a novel pipeline for personalized task synthesis. The process begins with heuristically prototyping instruction skeletons from tool invocation chains and API documentation. This is followed by context injection, decoupling and fuzzification which transform these abstract skeletons into situated personal tasks. All synthesized data are then subject to rigorous manual review and revision to guarantee quality and diversity

**Tool Invocation Chain Sampling.** To ensure a wider spectrum of task diversity and complexity, we first generate prototype instructions by sampling tool invocation chains, rather than asking human annotators to compose tasks from scratch. Our topological sampling method is designed to satisfy five key considerations: (1) *Dependency*: To preserve inherent dependencies, tool chains are sampled at the unit level, defined as a combination of several tools, rather than as individual tools. In practice, MCP tools from the same server are often designed to be used sequentially, such as listing available models before invoking a specific one for usage. (2) *Personalization*: Each sampled chain must include at least one personalized tool. (3) *Deduplication*: We ensure that no two sampled tool chains are identical, maximizing the diversity of the generated dataset. (4) *Co-*

*herence*: The sampled chain must be semantically coherent, meaning the outputs of upstream tools must serve as valid and necessary inputs for downstream tools. (5) *Realism*: All sampled chains undergo a manual review process where any chain that does not reflect a realistic user scenario is discarded.

**Instruction Prototyping and Context Enrichment.** Using the curated set of high-quality tool chains $\mathcal{L}^*$ and their corresponding formal API documentation $\mathcal{D}$, we employ an LLM-based heuristic method to synthesize preliminary instruction templates, denoted as $S_{proto}$. Formally, $S_{proto} = \mathcal{H}(\mathcal{L}^*, \mathcal{D}, P)$, where specific entities (e.g., user identifiers, product names) are replaced with typed placeholders $P$ (e.g., *user_id*, *product_name*). This abstraction removes instance-specific details (Wu et al., 2024; Wang et al., 2024) while explicitly preserving parameter-level dependencies across tool calls and the global logical structure that orchestrates the tool chain. Since the resulting instruction skeletons lack contextual grounding, we further concretize them by injecting realistic content. Specifically, we randomly sample entity values from our constructed context-trees $\mathcal{C}$ and replace the placeholders in $S_{proto}$ accordingly, yielding instantiated instructions $S_{inst}$.

**Instruction Fuzzification and Context Decoupling.** To better resemble authentic user inputs, we deliberately introduce fuzziness by removing information that humans typically omit in natural instructions. We define *implicit context* as a set of underlying parameters that are essential for tool execution but frequently absent from user instructions (e.g., user_id of a coworker). Such information is often ambiguous or underspecified in natural language (Li et al., 2025), yet its intended values can be deterministically inferred from the environment state (e.g., locating the unspecified coworker by querying the information of a shared group). Our strategy explicitly separates implicit context $\mathcal{C}_{imp}$ from the natural intent instruction. Specifically, the fuzzy instruction is obtained by removing execution-specific parameters from the instantiated instruction, $S_{fuzz} = \mathcal{F}(S_{inst} \setminus \mathcal{C}_{imp})$, where $\mathcal{F}$ denotes a fuzzification operator.

**Rigorous Filtering and Human Verification.** Following the automated synthesis pipeline, the candidate data triples $(S_{fuzz}, \mathcal{C}_{total}, \mathcal{L}^*)$ undergo a multi-stage manual curation process to ensure benchmark fidelity and task quality. Annotators ensure strict consistency among the natural language instruction $S_{fuzz}$, the decoupled context $\mathcal{C}_{total}$, and the ground-truth tool chain $\mathcal{L}^*$, forming a coherent, solvable and unambiguous task. Moreover, we deliberately increase task difficulty through two primary strategies: increasing query quantities (e.g., from "summarize one post" to "summarize ten posts") and rigorously pruning necessary context by removing any information that can be procedurally derived via the available tool chain. Tool chains are syn-

chronously enlarged and evolved during this process. The human annotation guidance is provided in Appendix B.2.

### 3.4. Benchmark Evaluation

In MCP-Persona, we utilize both checkpoint- and execution-based evaluations to measure agent performances.

**Checkpoint-Based Evaluations.** This evaluation focuses on the intermediate values generated during the agent's multi-step execution. Checkpoints are established at the boundary between decomposed sub-tasks. Based on the comparison of model's input parameters and predefined standard input parameters and simulated tool output, a checkpoint is score independently based on LLM judges.

**Execution-Based Evaluations.** Within our simulated environments, we formally abstract each personalized tool as a simulated Create, Read, Update, Delete (CRUD) operation on the structured context file. Thus, we can derive an execution-based evaluation to directly measure the success rate of the CRUD operations. We categorize checkpoints into three classes: *Generic Search*, *Personalized Search* (involving only Read operations), and *Personalized Operation* (involving other state-altering operations). For the two search types, human annotators execute tools to obtain ground-truth responses, which are compared against the agent's outputs for scoring. For *Personalized Operations*, three dedicated executors: `CreateDataExecutor`, `UpdateDataExecutor` and `DeleteDataExecutor` are implemented. LLM judges then determine whether the model has completed corresponding tasks based on the actual updates of the context in the application sandbox. All ground-truth data, context indices, and specifications undergo meticulous human annotation and verification.

## 4. Experiments

### 4.1. Basic Setups (Details in Appendix A)

We apply the proposed MCP-Persona benchmark to diverse LLM agents for comparative evaluation. We report three metrics. (1) *Checkpoint Accuracy (Acc)* is defined as the average checkpoint score within a task, where each checkpoint is scored by the LLM judge. (2) *Success Rate at 0.8 (SR-0.8)* measures the proportion of tasks whose *Acc* exceed 0.8. (3) *Execution Accuracy (Exec-Acc)* is defined as the average executor-verified checkpoint correctness over all human-specified execution steps within a task. All columns except the last three under *Overall* report *Acc* in Table 3.

Our benchmark includes two types of tasks: *Single-Server* tasks, which use tools from one personalized server, and *Cross-Server* tasks, which require coordinating tools across multiple personalized servers. In both settings, tool chains may also include information-seeking servers such as Amap.

*Table 3.* Performance of LLM agents on MCP-Persona. We report *Acc* except for the last two columns. The results highlight a significant challenge: even leading models struggle, achieving less than 50% accuracy, with Claude-Sonnet-4.5 delivering the best performance.

| Model | Single-Server | | | | Cross-Server | | | Overall | | |
|---|---|---|---|---|---|---|---|---|---|---|
| | Collaboration Platform | Content Management | Social Media | Email | Lark-Centric | Rednote -Centric | Hodgepodge | Acc | SR-0.8 | Exec-Acc |
| *Proprietary Models* | | | | | | | | | | |
| A\ Claude-Sonnet-4.5 | 39.94 | 19.76 | 47.04 | 43.63 | **40.81** | **42.37** | 12.50 | **38.66** | **10.40** | **41.50** |
| § GPT-5 | **43.50** | **22.57** | 42.64 | 47.17 | 37.67 | 34.66 | 12.50 | 36.99 | 6.94 | 41.45 |
| A\ Claude-Opus-4.1 | 38.79 | 13.56 | 44.79 | 9.71 | 39.67 | 34.70 | 25.00 | 34.52 | 7.05 | 36.77 |
| § o4-mini | 34.38 | 21.22 | 35.61 | **53.83** | 30.43 | 25.25 | 6.25 | 30.70 | 5.78 | 34.73 |
| § o3 | 26.41 | 14.55 | 32.78 | 41.08 | 34.64 | 26.05 | **37.50** | 29.79 | 5.20 | 30.27 |
| § GPT-4o | 24.50 | 7.58 | 36.98 | 12.57 | 30.65 | 20.29 | 25.00 | 25.56 | 4.35 | 20.02 |
| Ø Grok-4 | 17.80 | 11.82 | 39.43 | 5.71 | 26.78 | 19.79 | **37.50** | 24.58 | 6.68 | 22.49 |
| G Gemini-3-Pro | 14.01 | 11.03 | 23.38 | 36.92 | 14.60 | 22.05 | 6.25 | 16.91 | 1.79 | 5.78 |
| G Gemini-2.5-Pro | 22.58 | 22.24 | 26.22 | 20.92 | 18.23 | 11.76 | 6.25 | 20.68 | 0.66 | 13.38 |
| *Open-Source Models* | | | | | | | | | | |
| ⅓ Qwen3-Max-Latest | 24.36 | 11.67 | **47.98** | 11.71 | 30.95 | 15.79 | 18.75 | 27.54 | 5.75 | 29.23 |
| ⅓ Qwen3-235B-A22B | 23.55 | 12.12 | 40.05 | 13.71 | 30.40 | 19.25 | 31.25 | 26.75 | 4.07 | 21.83 |
| ◈ DeepSeek-V3 | 19.22 | 11.48 | 38.35 | 30.79 | 27.91 | 18.89 | 18.75 | 25.29 | 3.47 | 27.52 |
| ⅓ Qwen3-Coder | 23.50 | 13.18 | 31.34 | 8.29 | 29.80 | 23.57 | 6.25 | 23.93 | 3.65 | 20.14 |

*Table 4.* Comparison of simulation fidelity on the Lark Server (50 samples). *Tool-Traverse* aligns significantly better with real-world tool behaviors than the documentation-only *Vanilla* baseline. TP/TN: True Positive/Negative; FP/FN: False Positive/Negative.

| Method | Confusion Matrix | | | | Behavioral Alignment (%) | | | | Response Similarity | | | |
|---|---|---|---|---|---|---|---|---|---|---|---|---|
| | TP | TN | FP | FN | Acc | Prec | Rec | F1 | TF-IDF | ROUGE | BLEU | METEOR |
| Vanilla | 12 | 17 | 8 | 13 | 58.0 | 60.0 | 48.0 | 53.3 | 0.2113 | 0.2258 | 0.2307 | 0.3214 |
| **Tool-Traverse** | **23** | **24** | **1** | **2** | **94.0** | **95.8** | **92.0** | **93.8** | **0.7391** | **0.7366** | **0.7412** | **0.8703** |

## 4.2. Performance Comparison on MCP-Persona

**Main Results.** (1) Across servers and settings, SOTA LLM agents still show limited reliability on MCP-Persona as none achieved an accuracy exceeding 50% in either checkpoint or execution evaluations. (2) Performances vary by tool family on single-server scenarios and degrades further with richer context and longer-horizon coordination. These results reveal common failure modes in tool use and demonstrate that our benchmark is a useful testbed for evaluating and improving agent capabilities.

**Single-Server Breakdown.** (1) Email tasks achieve the highest accuracy for most models due to simple operations and short dependency chains. (2) Tasks involving richer schemas or heterogeneous objects, such as social media and collaboration platforms, are more challenging, as they require handling cross-user interactions and implicit entity resolution. (3) Content-management tools perform worst, reflecting agents' limited robustness when navigating and editing long documents under extended context budget.

**Cross-Server Breakdown.** We evaluate three cross-server scenarios: *Lark-Centric*, *Rednote-Centric*, and a *Hodgepodge* scenario spanning arbitrary combinations of the available applications. The hodgepodge subset proves to be the most challenging, consistently yielding the lowest accuracy across most models. This difficulty is attributable to

its higher frequency of cross-server interactions and more complex dependency chains.

**Metric Comparison.** Overall, *Exec-Acc* is slightly higher than *Acc*, suggesting that human-annotated execution checkpoints are less noisy than LLM-segmented checkpoints. *SR@0.8* remains low, indicating extreme difficulty in completing tasks end-to-end.

## 4.3. Validation of Tool Simulation Fidelity

To justify the effectiveness of our Tool-Traverse paradigm described in Section 3.1, we quantitatively verify that our code-based simulated tools $\mathcal{T}_{sim}$ exhibit behaviors and responses indistinguishable from real-world tools $\mathcal{T}_{real}$.

**Setups and Context Reconstruction.** We evaluate all 14 tools from Lark using 50 authentic interaction traces (25 successful, 25 failed). A critical challenge in simulation is ensuring state consistency; a simulator might fail simply because it lacks a specific entity (e.g., *user_id*) present in the environment, not because of flawed logic. To eliminate this confounding variable, we employ a *Context Reconstruction* strategy: we reverse-engineer the exact pre-condition state from each real-world trace and inject it into the simulator. This guarantees that $\mathcal{T}_{sim}$ and $\mathcal{T}_{real}$ operate on the exact same state snapshot, ensuring a fair comparison based solely on execution logic. We compare our approach against

*Table 5.* Comparison with and without skill documentations on the Lark and Rednote subsets.

| Model | Lark Subset | | | | | | Rednote Subset | | | | | |
| | Vanilla | | + OpenClaw Skills | | + Our Skills | | No Skills | | + OpenClaw Skills | | + Our Skills | |
| | Acc | Exec-Acc | Acc | Exec-Acc | Acc | Exec-Acc | Acc | Exec-Acc | Acc | Exec-Acc | Acc | Exec-Acc |
|---|---|---|---|---|---|---|---|---|---|---|---|---|
| ⑤ GPT-5 | 37.50 | 64.29 | 42.50 | 71.43 | **45.00** | **80.36** | 42.19 | 31.25 | 35.94 | 25.00 | **43.75** | **31.25** |
| A\ Claude-Sonnet-4.5 | 27.70 | 69.64 | 29.70 | 73.21 | **33.00** | **78.57** | **31.25** | **18.75** | 28.13 | 17.19 | 28.13 | 15.63 |
| 🐋 DeepSeek-V3 | **20.80** | **22.50** | 15.00 | 42.86 | 16.60 | 49.60 | 26.56 | 7.81 | 28.13 | 23.44 | **31.25** | **23.44** |
| 🐋 Qwen3-235B-A22B | 20.20 | 37.30 | 21.30 | **49.80** | **29.30** | 43.10 | 25.00 | 18.75 | 28.13 | 15.63 | **35.94** | **23.44** |
| Ӄ Kimi-K2.5 | 31.40 | 69.20 | 30.40 | 77.70 | **34.10** | **78.30** | 40.63 | 39.06 | 40.63 | 32.81 | **42.19** | **40.63** |

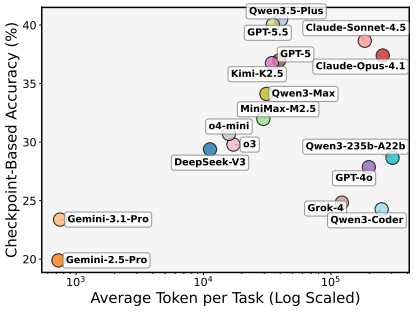

*(a)* Average token per task.

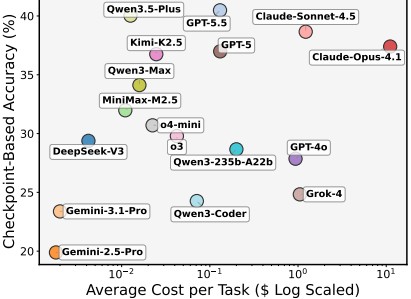

*(b)* Average cost ($) per task.

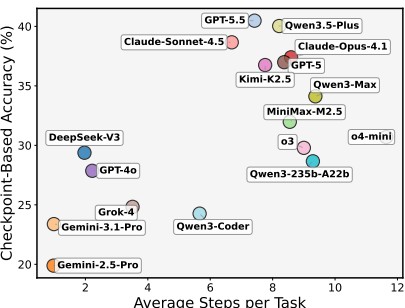

*(c)* Average step length per task.

*Figure 3.* Analysis of efficiency and performance trade-offs across various models, based on average token count, cost, and step length.

*Table 6.* Ablation study on the candidate tool settings.

| Model | All Tools | | Selected Tools | |
| | Acc | Exec-Acc | Acc | Exec-Acc |
|---|---|---|---|---|
| ⑤ GPT-5 | 41.04 | 29.25 | 40.46 | 46.23 |
| G Gemini-2.5-Pro | 20.23 | 13.68 | 22.54 | 11.79 |
| 🐋 Qwen3.5-Plus | 40.46 | 43.87 | 42.20 | 47.64 |
| ⅲ MiniMax-M2.5 | 32.37 | 34.43 | 36.42 | 38.21 |
| Ӄ Kimi-K2.5 | 36.99 | 38.21 | 39.88 | 37.26 |

a *Vanilla* baseline, which simulates tools relying only on documentation without behavioral traversal data.

**Results.** As shown in Table 4, (1) Tool-Traverse significantly outperforms the baseline on all metric families. The *Vanilla* baseline achieves only 53.3% F1, with a particularly high False Negative rate, indicating it struggles to handle valid but complex inputs due to hallucinated constraints. (2) On response similarity, Tool-Traverse roughly triples both TF-IDF and METEOR. These gaps directly reflect that the baseline emits generic error templates rather than the server's specific error codes and formats, whereas Tool-Traverse faithfully reproduces the nested response structure, a prerequisite for downstream agent reasoning to remain stable when run against $\mathcal{T}_{sim}$ in place of $\mathcal{T}_{real}$.

### 4.4. Ablation Study on Skills and Tool Candidates

We conduct additional experiments to validate whether skills enhance performance on MCP-Persona. Specifically, we compare the most downloaded OpenClaw skills obtained from ClawHub with a manually refined version (denoted as *Our Skill*). The skills are tailored for Lark and Rednote respectively, and are tested on their designated subsets. The refined version features more detailed descriptions of tool functionalities and parameters, thereby enabling the agent to leverage the available tools more effectively. The results are presented in Table 5. Overall, incorporating skill descriptions tends to improve agent performance. In particular, the manually refined skills further enhance performance compared with the OpenClaw version, although the magnitude of improvement varies across different models.

We also compare two tool candidate selection settings: (1) using all available tools and (2) using only tools from the ground-truth servers. As shown in Table 6, reducing the number of candidate tools improves performance, especially in cases with longer contexts.

### 4.5. Token and Cost Analysis

Figure 3 illustrates the relationship between monetary cost, token consumption, and checkpoint-based accuracy. Notably, Notably, GPT-5 stands out as the most cost-effective model, reaching 36.99 accuracy at only $0.09 per task on average, while Qwen3-Plus delivers the best trade-off among open-source alternatives. The results reveal no clear correlation between spending and performance, suggesting that model selection should prioritize accuracy-cost trade-offs rather than raw resource investment.

*Table 7.* Correlation analysis between human and LLM judgments, with misaligned ratios broken down by subset. Ckpt: Checkpoint.

| Task Category | Misaligned Ckpt | Total Ckpt | Ratio (%) | Task Category | Misaligned Ckpt | Total Ckpt | Ratio (%) |
|---|---|---|---|---|---|---|---|
| Lark_Short | 2 | 20 | 10.00 | Instagram | 2 | 29 | 6.90 |
| Lark_Long | 4 | 79 | 5.06 | Slack | 6 | 53 | 11.32 |
| Rednote_Short | 2 | 17 | 11.76 | Lark_Rednote | 11 | 119 | 9.24 |
| Rednote_Long | 3 | 59 | 5.08 | Lark_Obsidian | 4 | 50 | 8.00 |
| Notion_Short | 1 | 13 | 7.69 | Lark_Notion | 2 | 42 | 4.76 |
| Notion_Long | 2 | 24 | 8.33 | Lark_Other | 6 | 73 | 8.22 |
| Obsidian_Short | 1 | 17 | 5.88 | Rednote_Obsidian | 3 | 45 | 6.67 |
| Obsidian_Long | 3 | 21 | 14.29 | Rednote_Notion | 4 | 39 | 10.26 |
| Email | 7 | 51 | 13.73 | Rednote_Other | 7 | 81 | 8.64 |
| Wecom | 4 | 40 | 10.00 | Hodgepodge | 7 | 84 | 8.33 |
| Reddit | 1 | 14 | 7.14 | **Overall** | 82 | 970 | 8.45 |

## 4.6. Trajectory Error Analysis

By analyzing sampled failed traces, we have identified three recurring failure archetypes that account for most end-to-end failures in our tool-using setting.

**Under Exploration of Environment.** In many real-world scenarios, users do not specify all necessary details or preferences in their instructions. Faced with implied information, weaker models often fail by producing a superficially plausible action, without sufficiently exploring the environment or verifying constraints to uncover the missing information.

> **Task**: "...Additionally, please send a polite message to my supervisor, Song Ke, explaining my condition and requesting leave."
> **Context**: *Song Ke's user_id*: o9k5jtwo

The context contains Lark identity hints for Song Ke (i.e., user-id), but the instruction does not explicitly specify the platform. Instead of resolving Song Ke's identity and sending the message via Lark, the agent sends a WeCom message to a hallucinated recipient and then terminates. As a result, the output appears on-topic but fails the task due to platform mismatch and missing recipient grounding.

**Skipping Dependent Steps.** A second recurring failure is skipping latent dependency steps implied by tool schemas.

> **Task**: "...If everything looks fine, schedule the review meeting for next Monday from 10:00 AM to 12:00 PM in the main conference room. Also, have Zhao host the meeting since she...
> **Context**: *Zhao's phone number*: +86 13800138000

The intended workflow is to first resolve the platform-internal ID, *user_id*, from Zhao's phone number via the `user_batchGetId` tool. The resolved *user_id* is then passed to the `calendarEvent_create` tool to schedule a meeting with Zhao as the host. However, weaker models

often skip the resolution step, directly substituting the phone number for the *user_id* or fabricating an ID, which causes execution errors or silent premature termination.

**Over Long Context.** Our context-tree design can induce progressive context stacking across turns, and certain tools (e.g., local document readers) can return voluminous payloads that sharply increase the tightening the in-context window. As trajectories rollout, the model's ability to adhere to initial constraints and recall critical observations degrades, culminating in its failure to execute even rudimentary tasks.

## 4.7. Human-LLM Correlation Analysis

We conduct a correlation analysis based on checkpoint-level results of GPT-5 across all 173 tasks and 970 checkpoints. As shown in Table 7, the LLM-based judge is highly correlated with human judgment (91.5% alignment), largely due to our fine-grained breakdown of checkpoints. We also identify two primary causes for the remaining misalignments: (1) **Model Capacity Limitation**: Even advanced models occasionally struggle with complex logic or subtle context in specific tasks. (2) **Over-Strictness**: The judge model sometimes penalizes agents for using alternative tools, leading to a lower score despite successful execution.

## 5. Conclusion

In this work, we introduce **MCP-Persona**, a benchmark tailored for evaluating tool-augmented agents in realistic personalized settings. It is built upon 12 simulated MCP servers and 173 human-verified tasks, spanning a wide range of personal applications that previous work has long struggled to handle. Our experiments reveal that even SOTA agents fall short on personalized tool use, particularly in implicit grounding, multi-step state maintenance, and cross-tool coordination. We hope MCP-Persona serves as a reproducible, privacy-preserving testbed that accelerates progress on personalization-aware agents and ultimately bridges the remaining gap between intelligent assistants and the nuanced, context-rich workflows of real users.

## Impact Statement

We acknowledge that the development of benchmarks for personalized tool use, while intended to spur innovation, carries risks of societal impact, particularly in high-stakes domains.

1. A primary concern is the potential for such systems to perpetuate and amplify existing societal biases. For instance, a personalized tool in a domain like finance or healthcare, if trained on historical data reflecting systemic inequities, might learn to offer suboptimal recommendations or fewer opportunities to individuals from marginalized groups. This could lead to discriminatory outcomes in loan applications, medical diagnoses, or legal case preparation.

2. Second, the very act of "personalization" can create a risk of over-reliance and automation bias, where a user cedes their critical judgment to a system they perceive as being tailored to them. The personalization capabilities could be exploited by malicious actors for targeted phishing, social engineering, or automated spam.

3. Moreover, personalized agents that interact with sensitive services such as email, messaging platforms, or calendars raise significant privacy concerns. Improperly designed systems may expose personal data or introduce security vulnerabilities.

By creating a benchmark that prioritizes performance metrics like efficiency or task success rate, we risk incentivizing the development of powerful but inequitable models. Therefore, we stress that future research in this direction must be coupled with a rigorous focus on developing and integrating metrics for privacy, transparency, and accountability to mitigate these potential harms.

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

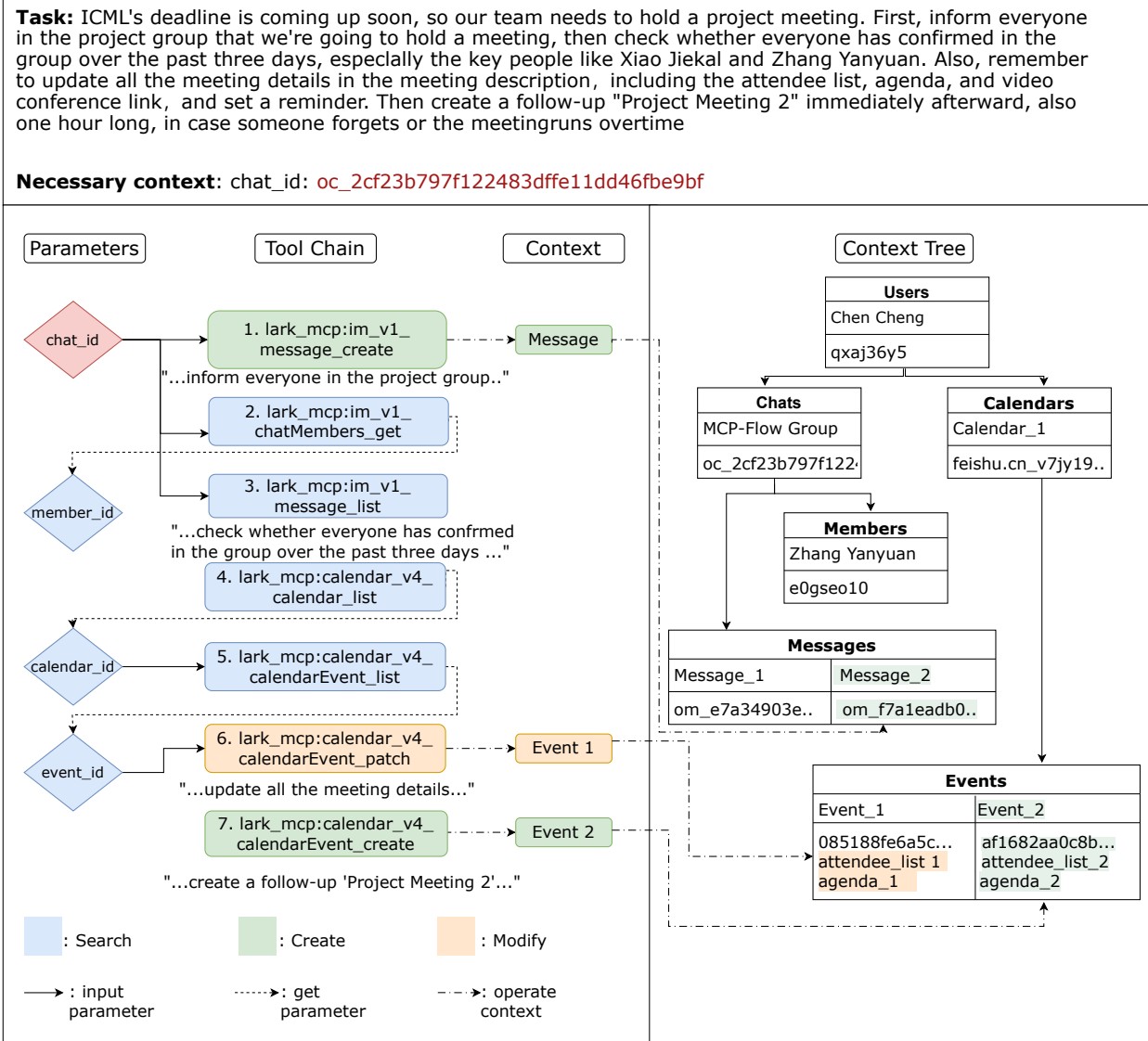

*Figure 4.* A visualized illustration of how tools and contexts interact in the simulated environments of MCP-Persona. Left: a representative Lark-MCP task where the agent progresses through a series of tool calls that either retrieve contextual information or apply state-modifying operations. Right: the corresponding Lark-MCP context-tree, the structured data accessed and updated by corresponding tools throughout the workflow.

## A. Implementation Details

### A.1. Tool and Data Details

**MCP Servers.** To ensure reproducibility and facilitate future research, we include the names and links of all MCP servers in our benchmark. Table 8 groups the MCP-Persona servers into generic search, social communication, enterprise collaboration, local note-taking, and email categories, along with their official repository URLs. Overall, these servers expose external tools and data sources to the model, enabling tasks such as retrieval, content management, and multi-step information gathering. Together they cover both personal information workflows and public information access that the benchmark evaluates.

*Table 8.* MCP servers included in the MCP-Persona Benchmark.

| MCP Server | Icon | URL |
|---|---|---|
| **Generic Search** | | |
| Amap | | `https://github.com/sugarforever/amap-mcp-server` |
| Mcpollinations | | `https://github.com/pinkpixel-dev/MCPollinations` |
| Context7 | | `https://github.com/upstash/context7` |
| Medical | | `https://github.com/JamesANZ/medical-mcp` |
| Baidu-maps | | `https://modelscope.cn/mcp/servers/@baidu-maps/mcp` |
| Monitor | | `https://github.com/seekrays/mcp-monitor` |
| Tushare | | `https://modelscope.cn/mcp/servers/@zhewenzhang/tushare_MCP` |
| Deepseek | | `https://github.com/DMontgomery40/deepseek-mcp-server` |
| Deep-research | | `https://smithery.ai/server/@baranwang/mcp-deep-research` |
| **Social Media** | | |
| Xiaohongshu | | `https://github.com/xpzouying/xiaohongshu-mcp` |
| Reddit | | `https://smithery.ai/server/reddit` |
| Instagram | | `https://smithery.ai/server/instagram` |
| Telegram | | `https://zapier.com/mcp/telegram` |
| Discord | | `https://github.com/hanweg/mcp-discord` |
| **Collaboration Platform** | | |
| Lark | | `https://github.com/larksuite/lark-openapi-mcp` |
| Slack | | `https://smithery.ai/server/slack` |
| Wecom | | `https://github.com/gotoolkits/mcp-wecombot-server` |
| **Email** | | |
| Gmail | | `https://zapier.com/mcp/gmail` |
| 163-Email | | `https://github.com/TimeCyber/email-mcp` |
| **Content Management** | | |
| Notion | | `https://github.com/suekou/mcp-notion-server?tab=readme-ov-file` |
| Obsidian | | `https://github.com/MarkusPfundstein/mcp-obsidian` |

## A.2. Experiment Details

**Main Experiments.** We evaluate 173 multi-modal tool-calling tasks spanning multiple MCP servers (e.g., Lark, Xiaohong-shu, Slack, Notion). Each task provides the agent with a user instruction, a set of available tools across information-seeking, personal file, and personal chat categories, and contextual information including necessary and unnecessary context. The agent is required to invoke tools in a multi-step loop (up to 20 rounds) to fulfill the user's request. We report three core metrics: (1) *Acc*, the average checkpoint-based evaluation score (0/0.5/1 per checkpoint, averaged over all checkpoints per task), (2) *Sr-0.8*, the proportion of tasks achieving a *Acc* score above 0.8, and (3) *Exec-Acc*, the execution-based evaluation score that checks final answer correctness and tool-calling parameter accuracy. All experiments use GPT-4o (Hurst et al., 2024) as the LLM judge, following previous research (Mo et al., 2025; Yuan et al., 2025; Du et al., 2026).

**Tool Simulation Fidelity.** For each trace, we replay the recorded input parameters against the simulated tools and compare the resulting response to the real one. Because boolean status flags are unreliable (a real server may return a wrapper-level success while embedding an error code in the payload), we use an LLM judge conditioned on the tool name, description, input schema, and full response body, explicitly prompted to assess whether the response substantively fulfills the tool's declared functionality. We use a low decoding temperature (0.1) for determinism, with a rule-based fallback over structured JSON error indicators when the judge output is unparseable. Treating real-server success as the positive class, we accumulate a confusion matrix where a *False Positive* (the most harmful failure mode) corresponds to the simulator silently accepting an invalid request, and a *False Negative* corresponds to over-rejection from hallucinated constraints.

We report two complementary metric families. (1) *Behavioral Alignment* measures whether the real tool and the simulated tool agree on the binary success/failure outcome: *Accuracy*, *Precision*, *Recall*, and *F1* computed from the confusion matrix

*Table 9.* The impact of context composition on model performance. This ablation study reveals that model performance is sensitive to distractor information in the context, a vulnerability that presents a clear opportunity for improvement.

| Model | Only Necessary Context | | With Context Distractors | |
|---|---|---|---|---|
| | Acc | Exec-Acc | Acc | Exec-Acc |
| GPT-5 | 41.04 | 29.25 | 36.99 | 39.15 |
| Gemini-2.5-Pro | 20.23 | 13.68 | 20.81 | 14.62 |
| Qwen3.5-Plus | 40.46 | 43.87 | 38.73 | 44.34 |
| MiniMax-M2.5 | 32.37 | 34.43 | 32.95 | 29.25 |
| Kimi-K2.5 | 36.99 | 38.21 | 38.73 | 40.57 |

above. F1 is the headline indicator since the 25/25 trace split is class-balanced and both error modes (*FP* and *FN*) carry distinct semantic costs as discussed above.

(2) *Response Similarity* measures how closely the simulated response payload reproduces the real one at the lexical and structural level. For each trace pair, we serialize both responses to canonical JSON strings and compute four standard text-similarity metrics: *TF-IDF*, which is sensitive to low-frequency tokens and therefore directly probes whether the simulator reproduces server-specific error codes and identifier formats; *ROUGE*, an overlap-based recall measure of the simulated response against the real one as reference, capturing whether the simulator preserves the key tokens of the real response; *BLEU*, n-gram precision (with brevity penalty), a stricter precision-oriented measure of phrase-level overlap; and *METEOR*, which augments unigram matching with stem/synonym matching and a chunk-level fragmentation penalty, making it the most sensitive of the four to whether the simulator reproduces the structural layout of the response (nested fields, error envelopes, identifier formats) rather than merely matching surface words. We average each metric across the 50 traces.

**Skills Ablation.** For the skills study in Section 4.4 and Table 5, we evaluate three settings on two domain-specific subsets: Lark (63 tasks) and Xiaohongshu/Rednote (64 tasks). The *Vanilla/No Skills* setting injects no additional skill document. The + *OpenClaw Skills* setting uses relevant public skills obtained from ClawHub [1]. The + *Our Skills* setting uses manually refined operational guides aligned with the MCP-Persona tool interfaces: for Lark, the guide is built around the Lark OpenAPI MCP server[2] and covers calendar, contact, and IM workflows; for Xiaohongshu, the guide is refined from the public Xiaohongshu MCP skill collection[3] and adapted to the benchmark's task workflows.

OpenClaw Skills can still contain operationally incorrect or stale guidance, such as tool inputs and outputs that no longer match the current API behavior, or instructions that emphasize setup rather than task execution. This ablation therefore tests whether skill documents help, and whether higher-quality, task-aligned procedural guidance is more beneficial than relevant but potentially inaccurate public skills.

**Tool Candidate Ablation.** For the candidate-tool study in Section 4.4 and Table 6, we compare *All Tools* and *Selected Tools*. In *All Tools*, all 140 tools from every MCP server are presented to the model, including 78 personalized tools for personal chat and file management. In *Selected Tools*, we retain only tools from the ground-truth servers required by the task. The relevant servers are identified from two human annotations: (1) task type labels (e.g., *feishu_single_long* denotes a single-server Lark task, while *xiaohongshu_notion* denotes a cross-server task), and (2) ground-truth tool-calling chains that specify the exact tools needed to complete the task.

**Context Confusion Ablation.** For the context-confusion study reported in Table 9, we compare *Only Necessary Context* with *With Context Distractors*. In the distractor setting, we append five additional paragraphs to the unnecessary context list while keeping the task instruction and necessary context unchanged. The distractors are randomly sampled from an entity pool comprising approximately 16k English paragraphs (40–100 words each), sourced from DBpedia (Mendes et al., 2012) entity descriptions and general-purpose text corpora. This tests the model's robustness to information overload.

---

[1] https://clawhub.ai
[2] https://github.com/larksuite/lark-openapi-mcp
[3] https://github.com/autoclaw-cc/xiaohongshu-mcp-skills

*Table 10.* A summary of the LLMs included in our study, detailing their versions or URLs.

| Model | Version |
|---|---|
| ⊛ GPT-5 (**Singh et al., 2025**) | `gpt-5-2025-08-07` |
| ⊛ o3 (**OpenAI, 2025**) | `o3-2025-04-16` |
| ⊛ o4-mini (**OpenAI, 2025**) | `o4-mini-2025-04-16` |
| ⊛ GPT-4o (**Hurst et al., 2024**) | `gpt-4o-2024-11-20` |
| ⊘ Grok-4 (**xAI, 2025**) | `grok-4-0709` |
| A\ Claude-4.5-Sonnet (**Anthropic, 2025b**) | `anthropic.claude-sonnet-4.5` |
| A\ Claude-4.1-Opus (**Anthropic, 2025a**) | `anthropic.claude-opus-4.1` |
| G Gemini-2.5-Pro | https://cloud.google.com/vertex-ai/generative-ai/docs/models/gemini/2-5-pro |
| G Gemini-2.5-Flash | https://cloud.google.com/vertex-ai/generative-ai/docs/models/gemini/2-5-flash |
| K Kimi-K2.5 (**Bai et al., 2026**) | https://www.kimi.com/ai-models/kimi-k2-5 |
| ⩜ Minimax-M2.5 | https://huggingface.co/MiniMaxAI/MiniMax-M2.5 |
| 🐦 Qwen3-Max (**Yang et al., 2025**) | `qwen-3-max-latest` |
| 🐦 Qwen3-235B-A22B (**Yang et al., 2025**) | https://huggingface.co/Qwen/Qwen3-235B-A22B-Instruct-2507 |
| 🐦 Qwen3-Coder (**Yang et al., 2025**) | https://huggingface.co/collections/Qwen/qwen3-coder |
| 🐳 DeepSeek-V3 (**Liu et al., 2024**) | https://huggingface.co/deepseek-ai/DeepSeek-V3-0324 |

*Table 11.* Representative task examples from MCP-Persona, featuring realistic, detail-rich, and challenging yet executable instructions grounded in real-world personal workflows.

| |
|---|
| **Example 1.** Task: Tomorrow at 9:00 a.m. we're holding the quarterly marketing strategy meeting, and I'm busy preparing the meeting materials right now. Please set a reminder for Li Minghui on Slack, and also record a voice prompt in nova's voice style and send it over. Then help me draft a simple meeting agenda, listing key topics like market analysis and the creative proposal. Oh, and make sure to check everyone tagged "Mingri Technology" and message them so that every single one of them receives the notice—don't miss anyone and end up delaying things. |
| **Example 2.** Task: Recently I've been preparing a research project on urban healthy walking and need to gather some basic information. First, please check what parks and medical institutions are within 1 km of Jianguo Road, and then find the most convenient way to get from my home to the nearest park. Update this route information in the research plan, and also look up recent findings in medical journals on the health benefits of urban walking. Finally, organize all of these materials and place them in the research reference resources for easy follow-up and consultation. |
| **Example 3.** Task: I'm planning to drive out and relax this weekend—I'm currently in Taiwan. Please check the most convenient driving route from here to Taipei 101, and also look for some good local specialty restaurants within 500 meters nearby. Also, first confirm that I'm still logged into my Xiaohongshu account "Guokr," so I won't have to log in on the spot when I want to post an update. Finally, remember to add my weekend plan to the end of the weekly plan note in Obsidian so it'll be easier to review later. |

### A.3. Model Details

As shown in Table 10, we present the versions of the LLMs used in our evaluation. The model set covers both proprietary and open-weight systems from major providers, enabling a broad comparison across different model families and deployment sources.

## B. Examples and Prompts

### B.1. Task Examples

Table 11 showcases representative tasks from MCP-Persona, illustrating how our Persona-Gen pipeline produces instructions that are realistic, detail-rich, and challenging yet executable. These tasks are grounded in authentic personal workflows and span diverse application domains, reflecting the complexity and ambiguity that real users encounter in everyday tool use.

## B.2. Human Annotation Guide

### B.2.1. TASK ANNOTATION GUIDE

---

**Instruction-ToolChain-Context Alignment Annotation Guide**

**Human Annotation Guide**
**1. Instructions-ToolChains-Context Alignment Annotation**
**Core Objective**
Ensure strict semantic consistency among the natural language instruction, decoupled context, and ground-truth tool chain, forming a coherent, solvable, and unambiguous personalized task.
**Key Steps**

1. **Task Selection:** Pick tasks with clear structure and logical tool chains from the candidate dataset, prioritizing those with linear dependencies.

2. **Instruction Optimization:**
   - Increase task difficulty (e.g., scale query quantities).
   - Refine the instruction structure to align with real usage scenarios.

3. **Context Refinement:**
   - Minimize necessary context: Remove information that can be derived via tool calls.
   - Filter unnecessary context: Retain only context relevant to the task.
   - Update context content to match the instruction scenario.

4. **Tool Chain Adjustment:**
   - Verify tool solvability: Check for missing parameters.
   - Evolve tool chains synchronously with instruction optimization.

5. **Consistency Check:**
   - Ensure the instruction intent can be fully achieved via the tool chain.
   - Confirm context provides all parameters required.
   - Delete tasks with misaligned instruction-context-tool chains.

---

### B.2.2. CHECKPOINT ANNOTATION GUIDE

---

**Execution-Based-GT Annotation Human Guide**

**Human Annotation Guide**
**2. Execution-Based-GT Annotation**
**Core Objective**
Annotate ground-truth (GT) data to support execution-based evaluation, enabling accurate assessment of the agent's ability to complete personalized tasks.
**Annotation Categories & Requirements**
GT is formatted as a list of checkpoints, with four checkpoint types:

1. **Generic Search:**
   - Definition: Web searches using non-personalized tools (e.g., map search, medical data query).
   - Annotation Requirement: Provide the exact GT value (e.g., search results, calculated data).

2. **Personalized Search:**
   - Definition: Search operations using personalized tools (e.g., viewing calendar, accessing notes, checking social media posts).

---

- Annotation Requirement: Provide the exact GT value (e.g., calendar event details, note content).

3. **Operate:**
   - Definition: State-altering operations using personalized tools (Create/Update/Delete), with changes reflected in the context.
   - Annotation Requirements:
     - `tool`: Specify the tool used for the operation.
     - `operate_type`: Mark as "create", "delete", "modify", or "other".
     - `summary`: Describe the operation in natural language.
     - `context`: Provide the ID of the context affected by the operation.

4. **Other:**
   - Definition: Tasks that fall outside the above categories.
   - Annotation Requirement: Fall back to Log-based Evaluation, with no additional GT value required.

**Annotation Principles**

- Ensure GT values are accurate and reproducible.

- Align checkpoints with the tool chain sequence.

- Maintain consistency between GT annotations and the environment context.

## B.3. Prompt Templates for Executable Code-Based Tool Simulation

### B.3.1. DYNAMIC CONTEXT HANDLER GENERATION

**Dynamic Context Handler Generation Prompt**

**System Message**
You are a Python code generator for dynamic context handler with nested dictionary structure. Generate a Python module that provides CRUD operations for dynamic context management.

**User Message**
**CONTEXT STRUCTURE:**

- **Context format:** `Dict[str, Dict[str, Any]]`.

- **Context file name:** {context_file_name}

- **Entity paths:** {entity_paths JSON}

- **ID fields:** {id_fields JSON}

- **Parent relations:** {parent_relations JSON}

**CONTEXT SAMPLE (Structure Only):**
{context_sample JSON}

**REQUIREMENTS:**
**1. Module Functions (Must implement all):**

- `load_context(path)`: Dict.

- `save_context(path, data)`: Save context to JSON with indentation.

- `get_entity_by_path(data, path, id)`: Retrieve specific entities.

- `list_entities_by_path(data, path, filters, limit)`: Support wildcard `[*]` traversal.

- `create_entity_by_path(data, path, entity, id)`: Validate parent existence before creation.

- `update_entity_by_path(...)` / `delete_entity_by_path(...)`: Standard CRUD.

**2. Path Parsing Rules (CRITICAL):**

- Must handle IDs containing dots (e.g., `user.name@domain.com`).

- **Rule:** Parse brackets `[...]` FIRST to extract selectors, THEN split by `.` outside brackets.

- *Example:* `calendars[user.name].events` → Token 1: `calendars` / `user.name`, Token 2: `events`.

**3. ID Generation & Filtering:**

- Analyze context structure to generate appropriate IDs (e.g., UUID for events, email-like for calendars).

- Apply filters for wildcard paths; use path-IDs as strict filters for specific paths.

**4. Implementation Details (Parse Logic):** You must implement the `parse_path` function following this logic to ensure dot-containing IDs are handled correctly:

```
def parse_path(path: str) -> List[PathToken]:
    tokens = []
    i = 0
    while i < len(path):
        if path[i] == '.':
            i += 1; continue
        # Logic to find segment end, handling nested brackets
        # ... (Implementation details as specified)
        # Extract segment name and selector
        tokens.append(PathToken(name=name, selector=sel))
    return tokens
```

**OUTPUT RULES:**

- Generate ONLY the Python code.

- No markdown formatting, no explanations.

- The code must be ready to execute.

B.3.2. PROMPT TEMPLATE FOR SIMULATION KERNEL GENERATION

**Simulation Kernel Generation Prompt**

**System Message**
You are a Python code generator for API tool simulation with dynamic context. Generate a Python function that simulates the behavior of a real API tool using dynamic context operations.

**User Message**
**TOOL INFORMATION:**

- **Tool name:** {tool_name}

- **MCP name:** {mcp_name}

- **Description:** {description_short}

- **Input schema:** {input_schema_str}

{success_examples_text} {error_examples_text}

**DYNAMIC CONTEXT INFORMATION:**

- **Context structure:** Dict (single user) or List (multi-user).

- **Context file:** {context_file_name} located at `Path(__file__).parent.parent`.

- **Entity paths:** {entity_paths JSON}

- **ID fields:** {id_fields JSON}

- **Parent relations:** {parent_relations JSON}

- **Current entity:** Type={entity_type}, Path={entity_path}

- **Context Selection:** Read `CONTEXT_ID` from env vars ("all" or specific ID).

**CRITICAL REQUIREMENTS:**

**1. Function Structure:**

- Function name MUST be: `def analyze_response_patterns(parameters_used):`

- Return format: `{"success": bool, "error": str|None, "result": dict|None}`

**2. Dynamic Context Handler Integration:**

- **Import:** `from dynamic_context_handler import load_context, save_context, get/list/create/update/delete_entity_by_path`

- **Loading:** Load context at start. Select specific context based on `os.environ.get("CONTEXT_ID")`.

- **Persisting:** After ANY modification (Create/Update/Delete), you MUST call `save_context`.

**3. Tool Operation Logic:**

- **List Tools:** Use `list_entities_by_path`. Validate references first. Use large limit for pagination.

- **Create Tools:** Use `create_entity_by_path`. Validate parent entities exist (e.g., calendar_id).

- **Update/Delete Tools:** Validate entity existence before modification.

**4. Input & Reference Validation (CRITICAL):**

- Validate input format (required fields, types).

- **Reference Check:** Use `get_entity_by_path` to verify that referenced IDs (calendar IDs, folder IDs) actually exist in the current context context.

- Handle special keywords like "primary" by resolving them against the context.

**5. Response Format:**

- Success: Match the tool's expected JSON structure (from examples).

- Failure: Return specific error messages in the error field.

**OUTPUT RULES:**

- Generate ONLY the Python code.

- The code must be ready to execute without markdown formatting.

