# OpenReview forum: "MCP-Persona: Benchmarking LLM Agents on Real-World Personal Applications via Environment Simulation"
_ICML.cc/2026/Conference — ICML 2026 regular_

### Official Review · Reviewer_f6YB · 2026-03-13

**Soundness:** 3
**Presentation:** 3
**Significance:** 3
**Originality:** 3
**Overall Recommendation:** 3
**Confidence:** 3

**Summary:**

The authors intend to investigate the concept of tool-augmented LLM agents operating within highly personalized, stateful environments using the recently introduced Model Context Protocol (MCP). To this end, the paper introduces MCP-Persona, a novel benchmark designed to evaluate agents on real-world, personalized MCP tools such as social media platforms (Reddit, Xiaohongshu) and enterprise collaboration suites (Lark, Slack). To circumvent the privacy and reproducibility issues inherent to personalized APIs, the authors propose a "traverse-then-simulate" methodology called Tool-Traverse. This approach uses LLMs to synthesize executable Python mock tools (Code-as-Simulation) based on interaction traces collected from live servers. Furthermore, they develop Context-Tree to simulate structured user profiles and Persona-Gen to create a dataset of 173 human-verified tasks. The benchmark employs both checkpoint-based and execution-based evaluations. Experiments across various state-of-the-art proprietary and open-weight models reveal that current agents still struggle significantly with multi-step, personalized tool use, particularly in handling implicit contexts and long-horizon coordination.

**Compliance With Llm Reviewing Policy:**

Affirmed.

**Final Justification:**

I have seen the author’s response and efforts, but I am inclined to keep my score unchanged.

**Key Questions For Authors:**

See weaknesses.

**Limitations:**

The authors have not adequately discussed the limitations of their work.

1. It would be great to see a candid discussion about relying on LLMs as judges. Without comparing these scores to human judgments, it's hard to ignore the known evaluation biases these models bring to the table.
2. Rather than saying there are no consequences that must be highlighted, the paper really needs to address privacy risks. Giving autonomous agents access to personal emails, messaging platforms, and calendars is a massive security risk.
3. It would be helpful to touch on the potential for misuse, like bad actors using these highly personalized agents for targeted social engineering or automated spam campaigns.

**Strengths And Weaknesses:**

**Strengths:**

1. The paper addresses a critical gap by evaluating tool-augmented LLMs on the Model Context Protocol (MCP) in personalized, real-world applications (e.g., social media and collaboration platforms).
2. The proposed "Tool-Traverse" and "Code-as-Simulation" paradigms represent a substantial and innovative leap over static, rule-based mocking by utilizing LLMs to autonomously synthesize executable Python engines.
3. The empirical validation of the simulator is highly rigorous, successfully demonstrating a 93.8% F1 score in behavioral alignment against a 53.3% baseline.

**Weaknesses:**

1. While the execution-based evaluation is a commendable inclusion, the Checkpoint-based evaluation (Acc) still heavily relies on LLM judges to independently score intermediate values. The lack of a robust correlation analysis between these LLM scores and human judgments raises concerns about inherent evaluation biases.
2. The simulated environments rely on a synchronous, state-based Context-Tree, which contrasts with real-world social and enterprise platforms that are inherently asynchronous and highly dynamic. This turn-based sandbox may fail to evaluate an agent's robustness when handling concurrent external state changes or real-time interruptions.
3. Although the simulation framework is automated, the current benchmark only curates 7 personalized MCP servers, limiting the evaluation of broad generalizability. Furthermore, the "Persona-Gen" pipeline uses LLMs for instruction prototyping; despite manual review, this initial synthetic step risks injecting model-specific linguistic biases and might fail to capture the true diversity of idiosyncratic human requests.

---

> ### Author Rebuttal · Authors · 2026-03-30
>
> Thank you for recognizing the practically utility and novelty of MCP-Persona. We greatly appreciate the time and effort you dedicated to reviewing this paper. Below are our responses:
>
> ---
> &nbsp;
>
> ## Response to Weakness 1
>
> > W1: While the execution-based evaluation is a commendable inclusion, the Checkpoint-based evaluation still heavily relies on LLM judges... lack of a robust correlation analysis...
>
> Following your suggestion, we conducted a correlation analysis based on checkpoint-level results of GPT-5 across all 173 tasks and 970 checkpoints.
>
> [**Table E**. Human-LLM Correlation Analysis.]
> Task Category|Misaligned Ckpt|Total Ckpt|Ratio(%)|Task Category|Misaligned Ckpt|Total Ckpt|Ratio(%)
> --|--|--|--|--|--|--|--
> Lark_Short|2|20|10.00|Instagram|2|29|6.90
> Lark_Long|4|79|5.06|Slack|6|53|11.32
> Rednote_Short|2|17|11.76|Lark_Rednote|11|119|9.24
> Rednote_Long|3|59|5.08|Lark_Obsidian|4|50|8.00
> Notion_Short|1|13|7.69|Lark_Notion|2|42|4.76
> Notion_Long|2|24|8.33|Lark_Other|6|73|8.22
> Obsidian_Short|1|17|5.88|Rednote_Obsidian|3|45|6.67
> Obsidian_Long|3|21|14.29|Rednote_Notion|4|39|10.26
> Email|7|51|13.73|Rednote_Other|7|81|8.64
> Wecom|4|40|10.00|Hodgepodge|7|84|8.33
> Reddit|1|14|7.14|**Overall**|82|970|8.45
> _
>
> As shown in Table E, the LLM-based judge is highly correlated with human judgment (**91.5% alignment**), largely due to our fine-grained breakdown of checkpoints.
>
> We also identify two primary causes for the remaining misalignments:
> 1. Model Capacity Limitation: Even advanced models like GPT-4o occasionally struggle with complex logic or subtle context in specific tasks.
> 2. Over-strictness in Evaluation: The judge model sometimes penalizes agents for using alternative tools, leading to a lower score despite successful execution.
>
> ---
> &nbsp;
>
> ## Response to Weakness 2
>
> > W2: This turn-based sandbox may fail to evaluate an agent's robustness when handling concurrent external state changes or real-time interruptions.
>
> Thank you for raising this point.
> 1. We argue that MCP-Persona **primarily targets fairness and reproducibility** for research. Therefore, building an asynchronous sandbox is currently beyond our intended scope.
> 2. State changes have **minimal impact** in our setting: the agent has no direct access to the full database and only observes tool responses; impactful interruptions within a specific call are rare. We showcase an example at https://anonymous.4open.science/r/MCP-Persona-F85D/case.pdf.
> 3. State changes are not specific to MCP-Persona, but **a common challenge for real-world agents**. The issue is even more severe for web-search and GUI agents, where changes are more frequent and impactful. To the best of our knowledge, few prior studies explicitly address this concern.
>
> ---
> &nbsp;
>
> ## Response to Weakness 3
>
> > W3: Although the simulation framework is automated, the current benchmark only curates 7 personalized MCP servers...
>
> We thank the reviewer for the constructive feedback on the number of servers and respond as follows:
> 1. **Benchmark Expansion**: We incorporate 8 new servers (5 personalized: Discord, Telegram, Gmail, Obsidian, and WeCom; 3 information-seeking: US Weather, Exa, and Douban), effectively doubling the number of personal services which are central to everyday lives.
> 2. **Leading Scale and Uniqueness**: Table F shows that MCP-Persona's scale is competitive with top-tier benchmarks, featuring the most personalized servers to date. Moreover, we uniquely include 9 popular personal apps which are absent in existing benchmarks.
>
> [**Table F**. Server Scale Comparison.]
> Benchmark|Total|Personalized
> --|--|--
> AppWorld|9|8
> Tau-Bench|2|2
> MCP-Bench|28|1
> MCPMark|5|2
> MCP-Universe|11|1
> TOOLATHLON|32|3
> MCP-Persona|24|12
>
> > Despite manual review, this initial synthetic step risks injecting model-specific linguistic biases and might fail to capture the true diversity...
>
> We respectfully note that our tasks are highly diverse, because:
> 1. **Initial Tool Chain Diversity**: We exhausted combinations of more than 100 tools, laying the foundation for a diverse set of task candidates.
> 2. **Rigorous Human Filtering**: Prototype instructions serve only as hints. Annotators receive 5 hints per batch and select the best, unique ones; not all prototypes are used. If none are satisfactory, the entire batch is discarded. This rigorous process effectively corrects model-sepcific biases.
> 3. **Final Task Diversity**: The task embeddings visualized using 3 dimensionality reduction techniques (PCA, t-SNE, and UMAP) comprehensively prove diversity.
>
> Figures illustrating the tool chains and tasks are available at: https://anonymous.4open.science/r/MCP-Persona-F85D/figs/.
>
> ---
> &nbsp;
>
> ## Response to Limitations Section
>
> Thank you for the detailed suggestions. We will make sure to add a dedicated section to discuss these limitations.
>
> Please also **refer to our response to Reviewer q529**.
>
> ---
> &nbsp;
>
> Overall we hope we have addressed your concerns and will be grateful if you may consider increasing the score.

---

> > ### Author Rebuttal · Reviewer_f6YB · 2026-04-04
> >
> > Thanks you for the rebuttal and clarifications. I will keep my score

---

### Official Review · Reviewer_BAGM · 2026-03-13

**Soundness:** 3
**Presentation:** 3
**Significance:** 3
**Originality:** 3
**Overall Recommendation:** 4
**Confidence:** 3

**Summary:**

This paper introduces MCP-Persona, a novel benchmark designed to evaluate the personalized MCP invocation capabilities of large language models (LLMs). To construct this benchmark, the authors curated a dataset comprising seven social media/collaboration MCPs, two note-taking MCPs, and several information-seeking tools.
Then, the authors propose Tool-Traverse, which leverages LLMs to synthesize executable Python sandboxes that faithfully emulate real-world API behaviors; and the Context-Tree component, which is designed to maintain structured user profiles and stateful environments. . Experiments on 173 human-verified tasks across various LLMs, reveal that current LLMs still exhibit significant deficiencies in personalized tool utilization, thereby demonstrating the value of this work to some extent.

**Compliance With Llm Reviewing Policy:**

Affirmed.

**Final Justification:**

I decide to maintain the "weak accept" score, considering the authors' rebuttals and other reviewers comments.

**Key Questions For Authors:**

- How much human effort (e.g., in person-hours) involved in annotating the 173 tasks and constructing the Context-Trees for the those personalized servers?

**Limitations:**

--

**Strengths And Weaknesses:**

# Strengths
- The work focuses on an interesting problem that current MCP benchmarks lacking evaluating the stateful MCP tools (e.g., Slack and Lark) beyond generic information-seeking applications.
- The evaluation of personal agents is inherently challenging due to privacy concerns and the need for stable, authenticated user sessions. The "Code-as-Simulation" approach presented here offers a highly clever and pragmatic solution.
-  The implementation of "instruction fuzzification"—deliberately omitting explicit parameters, effectively simulates the implicit nature of authentic human usages.
# Weaknesses
- While the 173 tasks are human-verified and demonstrably of high quality, the overall dataset size is relatively modest when compared to existing benchmarks for generic tool use (e.g., ToolBench). This raises questions about the breadth of coverage for diverse personalized scenarios.
- The significant human effort required for annotating tasks, constructing Context-Trees, validating tool chains, and de-emphasizing contextual information raises concerns regarding the benchmark's long-term scalability and maintainability.

---

> ### Author Rebuttal · Authors · 2026-03-30
>
> Thank you for the time and effort you dedicated to reviewing this paper. We greatly appreciate your positive evaluation. Below are our responses:
>
> ---
> &nbsp;
>
> ## Response to Weakness 1
>
> > W1: While the 173 tasks are human-verified and demonstrably of high quality, the overall dataset size is relatively modest when compared to existing benchmarks for generic tool use (e.g., ToolBench).
>
> We appreciate the reviewer's recognition of the high quality of our human-verified tasks. The use of human annotation introduces an inherent trade-off between quality and quantity.
> 1. As MCP-Persona is positioned as the first real-world, personalized benchmark for tool-augmented agents, we **deliberately prioritize quality over quantity** to establish a reliable and authentic evaluation standard. This methodological choice distinguishes our work from benchmarks like ToolBench, which rely on purely LLM-generated data. While that approach bypasses the human verification costs, it can be susceptible to quality and fidelity issues.
> 2. We highlight that the initial size of our dataset is **comparable to, or surpasses many other prominent benchmarks** in the field, as substantiated by the statistics provided in Table B.
>
> [**Table B**. Task Scale Comparison with ICLR Benchmarks.]
> |Benchmark|Task Scale|
> --|--
> Tau-Bench|165
> MCP-Bench|104
> MCPMark|127
> MCP-Universe|231
> TOOLATHLON|108
> MCP-Persona|173
>
> ---
> &nbsp;
>
> ## Response to Weakness 2 and Question 1
>
> > W2: The significant human effort required for annotating tasks, constructing Context-Trees... raises concerns regarding the benchmark's long-term scalability and maintainability.
>
> The human effort involved in constructing MCP-Persona primarily stems from two stages: the collection of authentic seed function calls (FC) (Section 3.1.2) and task annotation. Other efforts are either standard across most benchmarks (e.g., MCP deploymeny), or involve minimal overhead (e.g., identifying the context hierarchy).
> 1. **Authentic FC Collection**: The time required for this stage varies; each call is obtained through a trial-and-error approach based on official MCP documentation. Normally, a valid FC can be secured within 5 minutes. For a new server, the estimated effort is calculated as:
> $5\text{ min} \times N_{\text{tools}} \times S$ where $N_{\text{tools}}$ is the number of tools and $S$ is the scale factor ($S≥1$, representing unique seed calls per tool).
> 2. **Task Annotation**: We employ a human-in-the-loop workflow. Annotators review, filter, and refine prototype instructions synthesized by the LLM. While this process is used to maximize data quality, it is **technically optional**, as the instructions post-filtering already meet the standards required for training.
>
> In summary, the human effort is **acceptable and scales linearly**: the main cost is seed FC collection (≈5–10 min per tool) plus light filtering. Task annotation is optional for higher quality, so long-term scalability and maintenance remain practical.
>
> [**Table C**. Human Effort Overview.]
> |Human-in-the-Loop|Required Effort|
> |--|--|
> |MCP Deployment|Standard Overhead|
> |Authentic FC|5-10 min / Tool|
> |Context Hierarchy|Minimal (Experienced)|
> |Task Filtering|Mimimal (1/5 Sample)|
> |Task Annotation|Detailed in Table D|
>
> > Q1: How much human effort (e.g., in person-hours) involved in annotating the 173 tasks and constructing the Context-Trees for the those personalized servers?
>
> To answer your question, we surveyed our annotators to collect time statistics during the annotation process. Please note that these figures are estimates provided by annotators based on the time elapsed for each batch of tasks; thus, there may be minor variances compared to precise timing.
>
> [**Table D**. Task Annotation Time (minute).]
> |Task Category|Time Estimated|Task Scale|Time Average|Task Category|Time Estimated|Task Scale|Time Average|
> --|--|--|--|--|--|--|--
> Lark_Short|150|9|16.7|Instagram|55|6|9.2
> Lark_Long|240|11|21.8|Slack|250|10|25.0
> Rednote_Short|45|10|4.5|Lark_Rednote|400|20|20.0
> Rednote_Long|85|10|8.5|Lark_Obsidian|100|8|12.5
> Notion_Short|40|6|6.7|Lark_Notion|70|6|11.7
> Notion_Long|60|4|15.0|Lark_Other|80|8|10.0
> Obsidian_Short|40|6|6.7|Rednote_Obsidian|95|8|11.9
> Obsidian_Long|50|4|12.5|Rednote_Notion|95|9|10.6
> Email|80|10|8.0|Rednote_Other|120|8|15.0
> Wecom|60|8|7.5|Hodgepodge|180|8|22.5
> Reddit|54|4|13.5|**Overall**|2349|173|13.6
>
> As shown in Table D, the average annotation time is about *14 minutes per sample and 40 person-hours in total*.
>
> Note that the reported time only covers annotation. Prior to annotation, we provided comprehensive guidelines (Appendix C) and conducted rigorous training sessions with hands-on guidance to ensure that all annotators adhered to our quality standards.
>
> ---
> &nbsp;
>
> Thank you again for your valuable feedback. These suggestions help us improve the clarity and completeness of our paper.
> We hope we have addressed your concerns and would be grateful if you may consider increasing the score.

---

> > ### Author Rebuttal · Reviewer_BAGM · 2026-04-04
> >
> > I will maintain the "weak accept" recommendation.

---

> > > ### Author Response · Authors · 2026-04-04
> > >
> > > Thank you for the comment. We are glad that we were able to address your concerns, and we sincerely appreciate your positive evaluation.
> > >
> > > Given that your concerns have been addressed, although we understand that you may prefer not to change the overall score, we would be grateful if you could consider increasing your confidence score.

---

### Official Review · Reviewer_q529 · 2026-03-17

**Soundness:** 3
**Presentation:** 3
**Significance:** 3
**Originality:** 3
**Overall Recommendation:** 4
**Confidence:** 4

**Summary:**

This paper presents MCP-Persona, a benchmark designed for evaluating agent performance on real-world, personalized MCP tools (e.g. RedNote, Slack, etc..). The authors manually created 7 MCP servers spanning critical categories, including enterprise collaboration platforms, social media, and content management. Experiments on 13 frontier LLMs demonstrate that all the agents have limited performance on this benchmark.

**Compliance With Llm Reviewing Policy:**

Affirmed.

**Key Questions For Authors:**

N/A

**Limitations:**

While the paper does include an impact statement section, the authors do not discuss the potential negative societal impact of this work. Personalized tool use in high-stake settings could bring many risks to individuals and society and the authors should discuss this carefully instead of arguing that this is purely machine learning research.

**Strengths And Weaknesses:**

Strength:
1. MCP-Persona presents a solid benchmark that is grounded in real personalized MCP tools
2. The tasks in the benchmark are verified by humans
3. The benchmark results suggest that all the models struggle with this task, demonstrating the necessity of this benchmark
4. Overall, this paper is well-written

Weakness:
1. It would be nice to discuss whether adding polished skills could help to improve the agents' performance.

---

> ### Author Rebuttal · Authors · 2026-03-30
>
> Thank you for recognizing the practically utility and novelty of MCP-Persona, and for the time and effort you dedicated to reviewing this paper. We greatly appreciate your evaluation. Below are our responses:
>
> ---
> &nbsp;
>
> ## Response to Weakness 1
>
> > W1: It would be nice to discuss whether adding polished skills could help to improve the agents' performance.
>
> We thank the reviewer for this valuable suggestion.
> Accordingly, we have conducted additional experiments by employing enhanced Lark skills on the Lark subset of MCP-Persona. We compare the most downloaded *OpenClaw Skill* sourced from ClawHub with our manually refined version (*Our Skill*), which features more detailed descriptions of tool functionalities and parameters.
>
> The results in Table A demonstrate that incorporating polished skills generally boosts agent performance. Specifically, **manual refinement further enhances these results**, although the performance gains vary considerably across different models.
>
> [**Table A**. Comparison with and without Skills on the Lark Subset.]
> |Model  | Vanilla |  | + OpenClaw  | Skills | + Our  | Skills |
> |---|---|---|---|---|---|---|
> | Metric  | Acc | Exec-Acc | Acc | Exec-Acc | Acc | Exec-Acc |
> | GPT-5 | 36.10 | 64.20 | 42.20 | 69.60 | 42.60 | 80.20 |
> | Claude-4.5-Sonnet | 27.70 | 69.20 | 29.70 | 72.90 | 33.00 | 77.10 |
> | DeepSeek-V3 | 20.80 | 50.20 | 15.00 | 42.10 | 16.60 | 49.60 |
> | Qwen3-235B | 20.20 | 37.30 | 21.30 | 49.80 | 29.30 | 43.10 |
> | Kimi-K2.5 | 31.40 | 69.20 | 30.40 | 77.70 | 34.10 | 78.30 |
>
> ---
> &nbsp;
>
> ## Response to Limitations Section
>
> > While the paper does include an impact statement section, the authors do not discuss the potential negative societal impact of this work. Personalized tool use in high-stake settings could bring many risks to individuals and society and the authors should discuss this carefully instead of arguing that this is purely machine learning research.
>
> We apologize for the insufficient detail regarding the potential social impact of our work in the original submission.
>
> To thoroughly address your suggestion and that of Reviewer f6YB, we have **revised and expanded the "Impact Statement" section**, elaborating on three aspects: *social bias, misuse, and privacy.*
>
> ```
> We acknowledge that the development of benchmarks for personalized tool use, while intended to spur innovation, carries risks of societal impact, particularly in high-stakes domains.
>
> 1. A primary concern is the potential for such systems to perpetuate and amplify existing societal biases. For instance, a personalized tool in a domain like finance or healthcare, if trained on historical data reflecting systemic inequities, might learn to offer suboptimal recommendations or fewer opportunities to individuals from marginalized groups. This could lead to discriminatory outcomes in loan applications, medical diagnoses, or legal case preparation.
>
> 2. Second, the very act of "personalization" can create a risk of over-reliance and automation bias, where a user cedes their critical judgment to a system they perceive as being tailored to them. The personalization capabilities could be exploited by malicious actors for targeted phishing, social engineering, or automated spam.
>
> 3. Moreover, personalized agents that interact with sensitive services such as email, messaging platforms, or calendars raise significant privacy concerns. Improperly designed systems may expose personal data or introduce security vulnerabilities.
>
> By creating a benchmark that prioritizes performance metrics like efficiency or task success rate, we risk incentivizing the development of powerful but inequitable models. Therefore, we stress that future research in this direction must be coupled with a rigorous focus on developing and integrating metrics for privacy, transparency, and accountability to mitigate these potential harms.
> ```
>
> ---
> &nbsp;
>
> Thank you again for your valuable feedback and the positive rating. Overall we hope we have addressed your concerns and will be grateful if you may consider increasing the score.

---

> > ### Author Rebuttal · Reviewer_q529 · 2026-04-05
> >
> > I would still keep my score

---

### Decision · Program_Chairs · 2026-04-30

**Decision:**

Accept (regular)

**Comment:**

This paper introduces MCP-Persona, a benchmark for evaluating LLM agents on personalized, stateful MCP tools, an important application scenario that is largely missing from existing benchmarks. Reviewers appreciated the realism of the benchmark design, the human verification of tasks, and the practical simulation framework for privacy-preserving evaluation. The experiments provide clear evidence that current agents struggle substantially in these personalized environments, underscoring the benchmark’s value. Remaining concerns mainly concern benchmark scale, reliance on LLM judges, and simulation realism, but these do not outweigh the contribution. Overall, I recommend acceptance.